# The Comparison of Various Types of Health Insurance in the Healthcare Utilization, Costs and Catastrophic Health Expenditures among Middle-Aged and Older Chinese Adults

**DOI:** 10.3390/ijerph19105956

**Published:** 2022-05-13

**Authors:** Sha Chen, Zhiye Lin, Xiaoru Fan, Jushuang Li, Yao-Jie Xie, Chun Hao

**Affiliations:** 1Department of Medical Statistics, School of Public Health, Sun Yat-sen University, Guangzhou 510080, China; chensh95@mail2.sysu.edu.cn (S.C.); linzhiye_1@163.com (Z.L.); fanxr7@mail2.sysu.edu.cn (X.F.); lijsh58@mail2.sysu.edu.cn (J.L.); 2School of Nursing, The Hong Kong Polytechnic University, Hong Kong 999077, China; grace.yj.xie@polyu.edu.hk; 3Sun Yat-sen Global Health Institute, Institute of State Governance, Sun Yat-sen University, Guangzhou 510080, China

**Keywords:** China, health insurance, health utilization, catastrophic health expenditures, older adults

## Abstract

Rapid aging in China is increasing the number of older people who tend to require health services for their poor perceived health. Drawing on the China Health and Retirement Longitudinal Study (CHARLS) 2018 data, we used two-part model and binary logistic regression to compare various types of health insurance in the healthcare utilization, costs and catastrophic health expenditures (CHE) among the middle-aged and older adults in China. Compared with uninsured, all types of health insurance promoted hospital utilization rate (ranged from 8.6% to 12.2%) and reduced out-of-pocket (OOP) costs (ranged from 64.9% to 123.6%), but had no significant association with total costs. In contrast, the association of health insurance and outpatient care was less significant. When Urban Employee Medical Insurance (UEMI) as reference, other types of insurance did not show a significant difference. Health insurance could not reduce the risk of CHE. The equity in healthcare utilization improved and healthcare costs had been effectively controlled among the elderly, but health insurance did not protect against CHE risks. Policy efforts should further focus on optimizing healthcare resource allocation and inclining toward the lower socio-economic and poor-health groups.

## 1. Introduction

China is gradually becoming one of the most rapidly aging countries around the world. Chinese population aged 60 and over increased from 168 million (12.4%) in 2010 to 264 million (18.7%) in 2020 [1,2]. This drastic demographic change has undoubtedly added a heavy burden to the Chinese health and welfare system [3]. Since middle-aged and older adults are more susceptible to disease and utilize health care, families facing this condition are at higher risk for catastrophic health expenditures (CHE). In 2015, the incidence of poverty associated with heavy medical expenditures was high, at 44.1% [4]. Previous studies have found that common factors affecting healthcare utilization and costs among elder people included gender, education level, income level, health insurance, convenience of access, health condition and need, etc. [5,6,7]. Among them, health insurance could reduce out-of-pocket (OOP) costs for patients and is a key factor in promoting healthcare utilization. The health insurance aims to guarantee that everyone with the same health service needs has access to the equal health services. In China, health insurance coverage has been above 95% since 2011, which means that the goal of universal health coverage has been almost achieved [8].

Currently, there are four main types of basic health insurance in China, namely the Urban Employee Health insurance (UEMI), the New Rural Cooperative Health insurance (NCMI), the Urban Resident Health insurance (URMI) and the Urban and Rural Resident Health insurance (URRMI). The UEMI was launched in 1994 for urban workers, and the NCMI and URMI programs were launched in 2003 and 2007 respectively, with the NCMI for rural residents and the URMI for urban jobless residents [9]. However, the fragmentation of the health insurance system could lead to inequality of health services utilization [9,10]. Then the URRMI was launched in 2016, which was the integration of URMI and NCMI. The integration of health insurance is still in progress, and multiple basic insurance policies still coexist in China. The characteristics of four main types of basic health insurance were described in Table 1.

Equity and efficiency are key policy goals of health care research [11]. The impact of different insurance on healthcare and costs has long been a hot topic of interest. Unfortunately, the Chinese health insurance system still falls short in guaranteeing equity and efficiency [2,12,13]. Previous studies have found large differences in inequality of healthcare utilization by social health insurance, with UEMI having the greatest inequality, followed by URMI, and NCMI having the lowest inequality [6,14]. Healthcare inequity still exists in rural areas after the integration of URMI and NCMI. There is still a certain gap between the actual and the expected goal of URRMI [15]. But most of the studies suffered from small sample size, under-representation and poor timeliness. Moreover, the integration of urban and rural health insurance is accelerating, and further research is needed on the current health insurance reform.

Our study aims to explore the association of most recent various types of health insurance, healthcare utilization and costs among the middle-aged and older adults in China, and to examine whether different types of health insurance could reduce the risk of CHE occurrence.

## 2. Materials and Methods

### 2.1. Study Population

This study was based on the 2018 CHARLS dataset, which was a national and longitudinal study conducted by the National School of Development at Peking University. CHARLS was designed to collect a set of high-quality microdata representative of Chinese middle-aged and older households and individuals aged 45 and above. The purpose was to analyze the aging of Chinese population and promote interdisciplinary research on aging [16].

In short, CHARLS used a multistage probability-proportional-to-size sampling method, stratified by regions and then by urban districts or rural counties and per capita gross domestic product. It was representative of the country as a whole, which contained 150 counties and districts in 28 provinces. The first round of CHARLS data collection was conducted in 2011, and follow-up surveys were conducted in 2013, 2015, 2018 [17]. In the current analysis, we used data from the latest round of follow-up in 2018. Our sample excluded residents with missing information on key variables and having more than one kind of social health insurance. Overall, 15,936 participants from CHARLS were included.

### 2.2. Variable Specifications

#### 2.2.1. Dependent Variables

The dependent variables were as follows: (1) healthcare utilization: whether used outpatient service in the past 4 weeks and inpatient service in the past year, and (2) healthcare cost: total medical costs and OOP costs for outpatient service in the past 4 weeks and OOP costs for inpatient service in the past year. (3) catastrophic health expenditures (CHE): whether the participant’s household occurred CHE in the past year.

Total costs were defined as total healthcare costs before insurance reimbursement. OOP costs were those paid by respondents after insurance reimbursement. According to the WHO’s definition, a household occurs CHE when OOP spending on healthcare equaled or exceeded 40% of a household’s capacity to pay (CTP) [18]. CTP was defined as the total expenditure of the household minus the food-based household expenditure [19].

#### 2.2.2. Types of Health Insurance

There were six types of health insurance: uninsured, UEMI, URMI, NCMI, URRMI and other insurance. Among other insurance included government medical insurance, medical aid, private health insurance, urban non-employed persons’ health insurance, long-term care insurance and so on. The government medical insurance was different from the four types of social health insurance. It mainly targeted at retired cadres, civil servants of a certain level, cadres of party and government agencies, and students in school. It had a broader scope of reimbursement and a higher reimbursement rate compared to social health insurance, even UEMI [20].

#### 2.2.3. Covariates

We included covariates according to the Andersen’s behavioral model, which was the classic model for studying and analyzing health service utilization, mainly applied to health system evaluation and health service research [21]. Andersen’s behavioral model denoted usage of health services was determined by predisposing factors, enabling factors and need factors: (1) predisposing factors included age (years), gender (male/female), marital status (married/others), education level (primary school and below/secondary school/college and above), retirement status (retired/not retired), social activity (participated/not participated), and health-related behaviors including smoking (yes/no), drinking (yes/no) and physical activity (yes/no). (2) enabling factors included area of residence (western China, central China, eastern China), household expenditure per capita, residents (urban residents/rural residents/rural migrants). We defined four groups on the basis of quartiles of household expenditure per-capita (quartile 1, <850.6 USD; quartile 2, 850.6 USD to <1592.3 USD; quartile 3, 1592.3 USD to <2895.2 USD; and quartile 4, ≥2895.2 USD). (3) health need factors, including with any chronic diseases (no/one/two or more), any limitations in activities of daily living (ADL) (yes/no) and instrumental activities of daily living (IADL) (yes/no) and self-rated health status (very good, good/fair/poor, very poor). ADL limitations indicated any self-reported difficulty in the following activities of daily living as yes: eating, bathing, dressing, getting up and using toilet. IADL limitations indicated any self-reported difficulty in the following activities as yes: doing housework, cooking, shopping, making phone calls, taking medication and managing money. ADL and IADL reflected the state of physical function.

All variables could be found in detail on the CHARLS website. (https://charls.charlsdata.com/pages/Data/2018-charls-wave4/zh-cn.html, accessed on 24 September 2020).

### 2.3. Statistical Analysis

Descriptive analyses were conducted to describe the background factors based on the Andersen’s behavioral model and healthcare utilization of subgroups and overall study population with different types of health insurance. We reported the frequency and percentage for each categorical variable and the mean and standard deviation for each continuous variable. Then we used the Kruskal–Wallis test to assess differences of continuous variables and Chi-square test to assess differences of categorical variables among the groups.

This study used the Two-Part model (2PM) to analyze the association of health insurance with healthcare utilization and costs. The 2PM first used a logistic regression model that examined the probability of an individual using healthcare and having incurred any costs in the inpatient or outpatient visit. The second part determined the relationship between health insurance and healthcare costs by selecting an ordinary least squares (OLS) model, which was used for individuals who have utilized healthcare and incurred healthcare costs [22,23]. We used binary logistic regression analysis to explore whether insurance could reduce CHE.

We used Y to denote the medical cost of a person, which takes values in the range [0, +∞). The presence or absence of medical costs is denoted by the indicator variable Z.
(1)$$Z=\{\begin{array}{c} 0\;\;\;\mathrm{if}\;\;\;Y=0\\ 1\;\;\;\mathrm{if}\;\;\;Y>0 \end{array}$$

A logistic regression model can be used to express the probability of the event Z as
(2)$$\mathrm{Pr}\left(Z=1\right)=\mathrm{Pr}(Y>0|X)=\frac{\exp\left({X\beta }_{1}\right)}{1+\exp\left({X\beta }_{1}\right)}$$

X is a set of possible covariates related to the probability of event Z, β_1_ is the model parameter to be estimated, and the subscript 1 denotes the first part of the two-part model. Pr (Z = 0) =1 − Pr (Z = 1).

Equation (2) can also be expressed in terms of the logit function as
(3)$$\mathrm{logit}\left(P\right)=\log\left[\frac{\left(\mathrm{Pr}\;\left(Z=1\right)\right)}{1-\mathrm{Pr}\left(Z=1\right)}\right]={\;\beta }_{1}X$$

The Ordinary Least Squares Model (OLSM) is used in this study, and the log-linear model can be written in the following form.
(4)$$E\left[\log\left(Y|Y>0,X\right)\right]={\;\beta }_{2}X\;$$

E[log(Y|Y>0,X)]  represents the conditional log-expectation level of a person with medical expenses. The model indicates that the log-expectation is linearly related to a set of covariates X.

The two independent models can be combined to construct a unified likelihood function, and the form of the likelihood function is
(5)$$L\left(\beta_{1},{\;\beta }_{2}\right)={\Pi L}_{i}\left(\beta_{1},{\;\beta }_{2}\right)$$

The maximum likelihood estimation method or constrained maximum likelihood estimation method is used to find the parameter estimates in the model. Since an individual with medical cost Y > 0 provides information for both the first model and the second model, the correlation between the two models needs to be considered. Under the two-part model, the unconditional expected medical cost of an individual with characteristics can be expressed as
(6)E(Y>0|X)=Pr(Z=1|Y>0)×E[log(Y|Y>0,X)]

We used the individual sample weights to produce population representative estimates and clustered the standard errors at the community level in exploratory analysis. Average marginal effects (probabilities) were reported for logit models, whereas coefficients were reported for OLS models. The statistical significance was defined as *p* < 0.05. Data analyses were conducted by using survey commands with STATA 16.0.

## 3. Results

### 3.1. Population Characteristics

Table 2 summarized the description of the variables. This study is based on a sample of 15,936 respondents aged 45 years and over. 1972 (12.4%), 597 (3.7%), 10,639 (66.8%), 1985 (12.5%) and 262 (1.6%) were enrolled in UEMI, URMI, NCMI, URRMI and other insurance respectively. 481 (3.0%) had no insurance. The mean age of participants was 62.3 years. 7478 (46.9%) participants were male, 13,606 (85.4%) were married and 5694 (35.7%) were retired. 10,638 (66.8%) participants had primary education or below, and only 288 (1.8%) had college education or above. 8425 (52.9%) participated in social activities, 4306 (27.0%) currently smoked, 4184 (26.3%) currently drank and 14,369 (90.2%) had exercised every week. 11,489 (72.1%) participants were rural residents, 2628 (16.5%) were urban residents and 1819 (11.4%) were rural migrants. Participants were equally distributed among the eastern (34.1%), central (33.2%) and western regions (32.7%). 9087 (57.0%) had two or more chronic diseases, 3755 (23.6%) had one disease, 2559 (16.1%) had any ADL and 3619 (22.7%) had any IADL. Almost half participants’ self-reported health was fair (47.7%), followed by poor/very poor (28.8%), very good/good (23.5%).

Furthermore, the samples were divided into six subgroups according to the types of insurance. The result in Table 2 showed that participants in various insurance significantly differed in above mentioned predisposing, enabling and need factors.

### 3.2. Health Care Utilization and Costs

Table 3 presented information about health care utilization, costs of outpatient and inpatient care services. In the total sample, 2466 (15.5%) received outpatient care in last month and 2480 (15.6%) received inpatient care in last year. In terms of outpatient costs, the mean of outpatient total costs and outpatient OOP costs were 195.0 RMB (29.5 USD), 126.1 RMB (19.1 USD). The mean of the outpatient reimbursement rate was 18.4%. In terms of inpatient costs, the mean of inpatient total costs and inpatient OOP costs were 2333.7 RMB (352.9 USD), 1210.0 RMB (183.0 USD). The mean of inpatient reimbursement rate was 47.5%. 2051 (12.9%) of the participants’ households incurred catastrophic health expenditures. The reimbursement rates of outpatient care for participants with insurance were in the range of 14.4% to 41.7%. The reimbursement rates of inpatient care for participants with insurance were in the range of 44.6% to 59.2%. The reimbursement rate of inpatient care was much higher than the reimbursement rate of outpatient care. The incidence of CHE ranged between 8.8% and 15.8%.

When compared with uninsured group, those with UEMI and other insurance tended to have higher total costs and reimbursement rate, and they were less likely to incur CHE.

### 3.3. Two-Part Model

Table 4 presented the results of the two-part model for healthcare utilization and health cost of outpatient services and inpatient services. We set uninsured and UEMI as reference respectively to see the difference in comparison results. All variables in Table 2 had been adjusted and reported in Appendix A.

For outpatient care, the magnitude of increase in using outpatient care for NCMI, URRMI and other insurance participants was 4.3%, 4.3% and 11.1%, respectively, as compared to the uninsured. As for outpatient total costs, only UEMI significantly increased total costs by 73.1% compared to the uninsured. Turning to the outpatient OOP costs, only other insurance reduced OOP costs of outpatient care by 108.5%. When UEMI as reference, only uninsured group showed significant difference. The results of uninsured participants were consistent with previous results. When the reference group was different, the signs of the results were opposite.

For inpatient care, any kind of insurance was associated with a higher likelihood to use inpatient care as compared to the uninsured. The magnitude of increase in using inpatient care for UEMI, URMI, NCMI, URRMI and other insurance was 12.2%, 11.5%, 8.6%, 8.6% and 9.3% respectively compared to the uninsured. While all insurance had no significant association with total costs of inpatient care. Turning to the inpatient OOP costs, having any kind of insurance was significantly associated with lower OOP costs as compared to the uninsured. UEMI, URMI, NCMI, URRMI and other insurance reduced OOP costs by 97.6%, 68.6%, 85.1%, 64.9% and 123.6% respectively. When UEMI as reference, only uninsured group and NCMI showed significant difference. Participants with NCMI were 3.6% less likely to use inpatient care, and the inpatient total costs were reduced by 3.26% compared to UEMI.

### 3.4. Catastrophic Health Expenditures

Table 4 presented the results of logistic regression for CHE. We also set uninsured and UEMI as reference respectively to see the difference. After adjusting for covariates, we found that none of any types of insurance could reduce CHE in two different comparisons. Appendix A showed CHE was more likely to occur in people with lower per capita household expenditure, two or more chronic diseases, ADL limitation and self-reported health status of fair or below.

## 4. Discussion

Based on 15,936 Chinese adults aged 45 years and older, this study compared the association of various types of health insurance with healthcare utilization, costs and CHE. This study found that all types of health insurance, especially UEMI, promoted utilization of hospitalization and reduced corresponding OOP costs, but had no significant association with total hospitalization costs. In contrast, health insurance had limited association with outpatient care utilization and costs. When compared to UEMI, other types of insurance did not show a significant difference, only participants with NCMI tended to use less inpatient care and spend less on total costs. We also found CHE could not be reduced by any type of insurance.

The UEMI, as the most generous health insurance, had the higher utilization rate and reimbursement rates for both outpatient and inpatient service, but this also leaded to higher medical costs. UEMI was for the group with stable jobs and better accessibility to health services, which had relatively higher funding criteria and reimbursement rates in comparison with other types of insurance [6,10]. Previous studies have shown higher reimbursement rates for URMI than for NCMI [6,10,24]. But we found that the differences in health care utilization and reimbursement rates for URMI, NCMI and URRMI were relatively small. In the exploratory analysis, when we took UEMI as reference, only NCMI showed significant decrease in inpatient care utilization and total costs besides the uninsured. Due to lack of health resources and poor economic conditions, rural residents had greater barriers to health care utilization than urban residents [24].

Consistent with previous research, we found health insurance had stronger association with the utilization of inpatient care but limited association with outpatient care among the elderly [2]. Understandably, inpatient care was preferred due to the high reimbursement rates and full medical services, while insurance coverage for outpatient care was inadequate and reimbursement rates were low [25]. Another reason was that Chinese primary health system was still inadequate, with medical resources flocking to large urban hospitals [10]. This situation was related to a fact that free-market economy became the central principle pushing many areas of public policy reforms in China [26]. The middle-aged and elderly people distrusted primary care and wanted better medical service. Inpatient care could provide more complete examinations, and hospitals could also get higher profits. There was an inducement to excessive medical treatment. With hierarchical medical system still being promoted, improvements to outpatient services in the primary health care system should be increased to prevent wasting medical resources [2,27,28].

Our findings contrasted with evidence from previous studies which showed having health insurance may lead to higher total medical costs [6,10]. We only found an increase in total outpatient costs due to UEMI. This might have something to do with the fact that UEMI has higher reimbursement rate. The same situation has been reported in other countries, such as Germany [29] and Thailand [30], where the most generous social health insurance tends to incur the highest charges for the same types of medical conditions. It has been found that insurance has no financial protection for households. Insurance could not reduce or even increase medical OOP costs [10,31]. However, some recent studies found that health insurance have been associated with boosting healthcare utilization and lower medical OOP costs [32,33,34], and our study supported the results. In detail, regarding inpatient costs, a study from India [34] found that social health insurance was significantly associated with decreasing inpatient OOP expenditures, consistent with our findings. As for outpatient costs, we concluded that social health insurance had little association with outpatient OOP expenditures, but conclusions in Turkey [35] and Ghana [36] were opposite. In recent years, Chinese medical reform policies have mainly focused on the reform of medical service prices, the encouragement of hierarchical diagnosis and treatment, and the encouragement of social medical services. China has launched a series of measures to control the unreasonable increase in medical costs [37,38]. The health insurance reform is achieved in the breadth of coverage in the population, the comprehensiveness of the benefits packages and increased reimbursement rates [39]. Our results may suggest that as Chinese health insurance system continues to improve, the equity in healthcare utilization has improved and health care costs have been effectively controlled. Health insurance’s induced spending effect due to the profit-driven nature of providers eased. This required more up-to-date research to support this conclusion.

A multi-country analysis showed most developed countries had advanced social institutions that protect households from CHE. Only the USA, Greece, Switzerland, and Portugal had more than 0.5% of households facing CHE [40]. The gap between China and developed countries was still quite large. Our results denoted that health insurance did not reduce the risk of CHE after adjusting for covariates, which was consistent with previous research [41,42,43]. China is still a developing country with a large proportion of low-income groups, which leaving social health insurance with little protection against CHE [44]. However, the findings differ from some of international studies that have shown that health insurance reduces the risk of CHE, depending on a number of factors, including different definitions of CHE [45,46], different thresholds [47,48,49], and differences in national circumstances [40]. Although health insurance could promote health services utilization by reducing OOP costs, the protective effect had been offset by the rapidly rising medical expenses and healthcare needs. What’s more, the reimbursement rates of health insurance were still inadequate for households with higher health service needs and lower economic levels [50]. As for the variables included according to the Andersen’s behavioral model, we also found participants with economically disadvantaged and poor health were more likely to have CHE [12,42,43]. It suggested that the policy of health insurance should be further improved to accurately identify the characteristics of the poor and incline toward the lower socio-economic and poor-health groups [43].

## 5. Strengths and Limitations

To the best of our knowledge, this is one of the first studies to explore the association of health insurance with healthcare utilization, costs and CHE simultaneously among the middle-aged and elderly population. We included almost all types of insurance in China in this study, which allowed for a comprehensive comparison of healthcare utilization across multiple groups.

Our findings had several limitations. First, we used self-reported survey data, which might suffer from recall bias and measurement error. Second, the study performed cross-sectional analyses, and no causal effects should be assumed. Third, our study findings are likely to be influenced by additional factors, which are not included in the claims data. These include severity of illness or patients’ social settings.

## 6. Conclusions

Our study found the gap between different insurance types is narrowing. While UEMI still had an advantage, different insurance could be effective in promoting health service utilization and lowering OOP costs. The equity in healthcare utilization improved and healthcare costs had been controlled among middle-aged and elderly adults, but health insurance did not protect against CHE risks. Our study had several suggestions for policymakers. First of all, the government should actively promote hierarchical medical system and primary care system, and reasonably allocate medical resources to avoid waste. Second, insurance integration has progressed, but it should continue to narrow the gap between different types of insurances. Reimbursement for outpatient services should be increased to reduce the waste of inpatient resources. Third, health insurance could not reduce the risk of CHE, but is mainly related to the financial situation and health status. The health insurance scheme should be tilted to socio-economic and poor-health groups and improve protection measures.

## Figures and Tables

**Table 1 ijerph-19-05956-t001:** The characteristics of four main types of basic health insurance [10].

	UEMI	NCMI	URMI	URRMI
**Date Started**	1994	2003	2007	2016
**Target population**	Urban employee	Rural residents	Urban residents without formal employment	Urban residents without formal employment and rural residents
**Enrolment**	Mandatory	Voluntary at household level but could be enforced once the county joins the NCMI	Voluntary	Voluntary
**Reimbursement rate, ceiling and deductibles**	Set by the city governments. The rates depend largely on the types of health providers	Set by the county government. The rates depend largely on the types of health providers	Set by the city government, but these rates are different for children, elderly and other urban residents. They also depend on the types of health providers	Set by the county governments. The rates depend largely on the types of health providers
**Covered services**	Inpatient services, catastrophic outpatient services, some prevention care services	Inpatient services, catastrophic outpatient services, some prevention care services	Mainly cover inpatient services and catastrophic outpatient services	Inpatient services and outpatient services

**Table 2 ijerph-19-05956-t002:** Background characteristics by predisposing, enabling and health need factors among older Chinese, 2018.

Variables	Total (N = 15,936) n (col%)	No Insurance (n_1_ = 481) n (col%)	UEMI (n_2_ = 1972) n (col%)	URMI (n_3_ = 597) n (col%)	NCMI (n_4_ = 10,639) n (col%)	URRMI (n_5_ = 1985) n (col%)	Other Insurance (n_6_ = 262) n (col%)	*p*-Value
**Predisposing factors**								
Age, mean (SD)	62.3 (9.9)	64.6 (11.9)	62.7 (10.1)	61.7 (10.0)	62.0 (9.7)	62.3 (9.8)	66.1 (10.7)	**<0.001**
Male	7478 (46.9)	204 (42.4)	1079 (54.7)	228 (38.2)	4913 (46.2)	914 (46.0)	140 (53.4)	**<0.001**
Married	13,606 (85.4)	331 (68.8)	1736 (88.0)	504 (84.4)	9140 (85.9)	1686 (84.9)	209 (79.8)	**<0.001**
Education								**<0.001**
Primary school and below	10,638 (66.8)	401 (83.4)	539 (27.3)	275 (46.1)	7879 (74.1)	1425 (71.8)	119 (45.4)	
Secondary school	5010 (31.4)	80 (16.6)	1221 (61.9)	305 (51.1)	2744 (25.8)	557 (28.1)	103 (39.3)	
College and above	288 (1.8)	0 (0.0)	212 (10.8)	17 (2.8)	16 (0.2)	3 (0.2)	40 (15.3)	
Retired	5694 (35.7)	195 (40.5)	1172 (59.4)	371 (62.1)	3174 (29.8)	623 (31.4)	159 (60.7)	**<0.001**
Having social activities	8425 (52.9)	217 (45.1)	1415 (71.8)	349 (58.5)	5236 (49.2)	1045 (52.6)	163 (62.2)	**<0.001**
Current smoking	4306 (27.0)	143 (29.7)	498 (25.3)	127 (21.3)	2978 (28.0)	499 (25.1)	61 (23.3)	**<0.001**
Current drinking	4184 (26.3)	108 (22.5)	601 (30.5)	135 (22.6)	2706 (25.4)	564 (28.4)	70 (26.7)	**<0.001**
Having physical exercise	14,369 (90.2)	395 (82.1)	1845 (93.6)	546 (91.5)	9582 (90.1)	1761 (88.7)	240 (91.6)	**<0.001**
**Enabling factors**								
Residents								**<0.001**
Urban residents	2628 (16.5)	37 (7.7)	1569 (79.6)	485 (81.2)	209 (2.0)	196 (9.9)	132 (50.4)	
Rural residents	11,489 (72.1)	401 (83.4)	244 (12.4)	44 (7.4)	9142 (85.9)	1557 (78.4)	101 (38.5)	
Rural migrants	1819 (11.4)	43 (8.9)	159 (8.1)	68 (11.4)	1288 (12.1)	232 (11.7)	29 (11.1)	
Per capital household expenditure								**<0.001**
Quartile 1 (~850.6 USD)	3986 (25.0)	163 (33.9)	66 (3.3)	63 (10.6)	3165 (29.7)	496 (25.0)	33 (12.6)	
Quartile 2 (850.6~1592.3 USD)	3983 (25.0)	120 (24.9)	270 (13.7)	127 (21.3)	2881 (27.1)	543 (27.4)	42 (16.0)	
Quartile 3 (1592.3~2895.2 USD)	3983 (25.0)	111 (23.1)	584 (29.6)	177 (29.6)	2530 (23.8)	509 (25.6)	72 (27.5)	
Quartile 4 (2895.2 USD)	3984 (25.0)	87 (18.1)	1052 (53.3)	230 (38.5)	2063 (19.4)	437 (22.0)	115 (43.9)	
Area								**<0.001**
West	5204 (32.7)	184 (38.3)	544 (27.6)	167 (28.0)	3686 (34.6)	553 (27.9)	70 (26.7)	
Central	5294 (33.2)	135 (28.1)	670 (34.0)	315 (52.8)	3623 (34.1)	469 (23.6)	82 (31.3)	
East	5438 (34.1)	162 (33.7)	758 (38.4)	115 (19.3)	3330 (31.3)	963 (48.5)	110 (42.0)	
**Health need factors**								
Any chronic disease								**<0.001**
No	3094 (19.4)	101 (21.0)	380 (19.3)	409 (20.6)	109 (18.3)	2052 (19.3)	43 (16.4)	
One	3755 (23.6)	135 (28.1)	386 (19.6)	446 (22.5)	134 (22.4)	2602 (24.5)	52 (19.8)	
Two or more	9087 (57.0)	245 (50.9)	1206 (61.2)	1130 (56.9)	354 (59.3)	5985 (56.3)	167 (63.7)	
With any ADL	2559 (16.1)	120 (24.9)	158 (8.0)	277 (14.0)	83 (13.9)	1879 (17.7)	42 (16.0)	**<0.001**
With any IADL	3619 (22.7)	191 (39.7)	186 (9.4)	413 (20.8)	107 (17.9)	2677 (25.2)	45 (17.2)	**<0.001**
Self-reported health status								**<0.001**
Very good/Good	3750 (23.5)	117 (24.3)	600 (30.4)	508 (25.6)	165 (27.6)	2298 (21.6)	62 (23.7)	
Fair	7599 (47.7)	206 (42.8)	1017 (51.6)	942 (47.5)	287 (48.1)	5009 (47.1)	138 (52.7)	
Poor/Very poor	4587 (28.8)	158 (32.8)	355 (18.0)	535 (27.0)	145 (24.3)	3332 (31.3)	62 (23.7)	

Abbreviations: UEMI, the Urban Employee Health insurance; URMI, the Urban Resident Health insurance; NRMI, the New Rural Cooperative Health insurance; URRMI, the Urban and Rural Resident Health insurance; ADL: activities of daily living limitations; IADL: instrumental activities of daily living limitations. Kruskal–Wallis test was used to assess differences of continuous variables and Chi-square test to assess differences of categorical variables among the groups.

**Table 3 ijerph-19-05956-t003:** Health care utilization and costs among older Chinese, 2018.

Variables	Total (N = 15,936) n (col%)	No Insurance (n_1_ = 481) n (col%)	UEMI (n_2_ = 1972) n (col%)	URMI (n_3_ = 597) n (col%)	NCMI (n_4_ = 10,639) n (col%)	URRMI (n_5_ = 1985) n (col%)	Other Insurance (n_6_ = 262) n (col%)	*p*-Value
**Outpatient care**								
Utilization in last month	2466 (15.5)	59 (12.3)	329 (16.7)	94 (15.7)	1642 (15.4)	302 (15.2)	40 (15.3)	0.290
Total cost, mean (SD), RMB	195.0 (1106.3)	161.5 (1165.6)	298.6 (1405.2)	166.5 (865.4)	176.3 (1033.8)	180.3 (985.8)	411.8 (2121.4)	0.110
OOP cost, mean (SD), RMB	126.1 (680.1)	161.5 (1165.6)	145.3 (668.3)	117.0 (494.8)	121.3 (668.5)	119.2 (592.9)	186.5 (953.9)	0.240
Reimbursement rate, mean (SD), %	18.4 (30.6)	0 (0.0)	35.8 (39.6)	14.4 (27.0)	15.3 (27.5)	18.1 (28.9)	41.7 (42.2)	**<0.001**
**Inpatient care**								
Utilization in last year	2480 (15.6)	41 (8.5)	385 (19.5)	108 (18.1)	1618 (15.2)	284 (14.3)	44 (16.8)	**<0.001**
Total cost, mean (SD), RMB	2333.7 (104,15.6)	1000.1 (6116.6)	4149.5 (14,935.8)	3409.2 (14,326.5)	1992.2 (9098.3)	2356.7 (11,057.7)	2361.5 (7890.0)	**<0.001**
OOP cost, mean (SD), RMB	1210 (6154.7)	1000.1 (6116.6)	1741.7 (7859.0)	1836.4 (8241.3)	1085.1 (5635.2)	1254.0 (6421.3)	916.2 (3284.2)	**<0.001**
Reimbursement rate, mean (SD), %	47.5 (29.0)	0.0 (0.0)	59.2 (27.5)	44.6 (26.1)	46.3 (28.4)	45.8 (28.2)	54.9 (31.2)	**<0.001**
**Catastrophic health expenditures**								
Yes	2051 (12.9)	76 (15.8)	178 (9.0)	70 (11.7)	1450 (13.6)	254 (12.8)	23 (8.8)	**<0.001**

Abbreviations: UEMI, the Urban Employee Health insurance; URMI, the Urban Resident Health insurance; NRMI, the New Rural Cooperative Health insurance; URRMI, the Urban and Rural Resident Health insurance; OOP: out-of-pocket; CHE: catastrophic health expenditures. Kruskal–Wallis test was used to assess differences of continuous variables and Chi-square test to assess differences of categorical variables among the groups. All costs are measured in Chinese RMB. 100 RMB = 15.12 USD.

**Table 4 ijerph-19-05956-t004:** Two-part model for health utilization and costs (inpatient and outpatient) and logistic regression for CHE.

Variables	Outpatient Care			Inpatient Care			CHE
	Utilization	Total Costs	OOP Costs	Utilization	Total Costs	OOP Costs	Logit
	Logit	OLS	OLS	Logit	OLS	OLS	
**Health insurance** (Uninsured as reference)							
UEMI	0.048 ^†^ (0.027)	**0.731 * (0.339)**	−0.211 (0.378)	**0.122 ** (0.019)**	0.475 ^†^ (0.273)	**−0.976 ** (0.302)**	0.014 (0.021)
URMI	0.047 (0.030)	0.428 (0.328)	0.165 (0.353)	**0.115 ** (0.024)**	0.356 (0.270)	**−0.686 * (0.303)**	0.033 (0.025)
NCMI	**0.043 * (0.017)**	0.259 (0.219)	−0.047 (0.226)	**0.086 ** (0.013)**	0.149 (0.230)	**−0.851 ** (0.244)**	0.004 (0.015)
URRMI	**0.043 * (0.020)**	0.307 (0.255)	−0.092 (0.259)	**0.086 ** (0.015)**	0.357 (0.251)	**−0.649 * (0.266)**	0.005 (0.017)
Other insurance	**0.111 ** (0.042)**	0.696 ^†^ (0.364)	**−1.085 * (0.509)**	**0.093 ** (0.028)**	0.532 ^†^ (0.282)	**−1.236 * (0.535)**	−0.006 (0.028)
**Health insurance** (UEMI as reference)							
Uninsured	−0.048 ^†^ (0.027)	**−0.731 * (0.339)**	0.211 (0.378)	**−0.122 ** (0.019)**	−0.475 ^†^ (0.273)	**0.976 ** (0.302)**	−0.014 (0.021)
URMI	−0.001 (0.023)	−0.303 ^†^ (0.180)	0.376 (0.250)	−0.007 (0.020)	−0.119 (0.140)	0.290 (0.204)	0.019 (0.022)
NCMI	−0.004 (0.025)	−0.472 ^†^ (0.281)	0.164 (0.313)	**−0.036 * (0.018)**	**−0.326 * (0.149)**	0.125 (0.183)	−0.009 (0.017)
URRMI	−0.005 (0.023)	−0.424 ^†^ (0.242)	0.120 (0.286)	−0.036 ^†^ (0.019)	−0.118 (0.153)	0.327 ^†^ (0.192)	−0.008 (0.018)
Other insurance	0.064 (0.051)	−0.035 (0.372)	−0.874 (0.663)	−0.029 (0.028)	0.058 (0.160)	−0.260 (0.444)	−0.020 (0.026)

Notes: Abbreviations: UEMI, the Urban Employee Health insurance; URMI, the Urban Resident Health insurance; NCMI, the New Rural Cooperative Health insurance; URRMI, the Urban and Rural Resident Health insurance; OOP: out-of-pocket; CHE: catastrophic health expenditures; ADL: activities of daily living limitations; IADL: instrumental activities of daily living limitations. Average marginal effects (probabilities) were reported for Logit models (logistic regression models), whereas coefficients were reported for OSL models (ordinary least squares models). Robust standard errors are reported in parenthesis. All variables in Table 2 had been adjusted and reported in Appendix A. ^†^ *p* < 0.10; * *p* < 0.05; ** *p* < 0.01; Statistically significant results were bolded.

## Data Availability

Data could be found in detail on the CHARLS website. (http://charls.pku.edu.cn/, accessed on 24 September 2020).

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
