# Peer review of "The Comparison of Various Types of Health Insurance in the Healthcare Utilization, Costs and Catastrophic Health Expenditures among Middle-Aged and Older Chinese Adults"

_ijerph, 2022, doi:10.3390/ijerph19105956_

Round 1
Reviewer 1 Report
It is an interesting study for the international audience.
I have the following comments for the authors:
-Is the problem of health care utilization for the elderly population, or is it mentioned here for the general population? It is not clear (lines 37-39).
-Line 42, what do you mean, stable? Not fully understood.
-What is the definition of catastrophic health expenditures? What is the percentage of it within the household income?
-The conclusion and discussion part of the manuscript does not specify how policy makers and practitioners can use these research results. There are also no specific policy recommendations. The results do not make the necessary discussions about providing health services for the elderly and the prevention of catastrophic health expenditure.
-The discussion section needs to be rewritten by comparing the research results with similar international studies. Looking at the references section, many of the references belong to China. It is proposed to enrich the discussion part by using other studies.
Author Response
It is an interesting study for the international audience. I have the following comments for the authors:
- Is the problem of health care utilization for the elderly population, or is it mentioned here for the general population? It is not clear (lines 37-39).
A: Sorry for the confusing. It’s for the elderly population. Now we have revised the expression in the text, line 38-40, page 1.
“Previous studies have found that common factors affecting healthcare utilization and costs among elder people included gender, education level, income level, health insurance, convenience of access, health condition and need, etc.”
- Line 42, what do you mean, stable? Not fully understood.
A: Sorry for the confusing. We wanted to express that health insurance coverage has been above 95% since 2011. We have revised the expression in the text, line 43-45, page 1-2.
“In China, health insurance coverage has been above 95% since 2011, which means that the goal of universal health coverage has been almost achieved.”
- What is the definition of catastrophic health expenditures? What is the percentage of it within the household income?
A: Thank you for your comments. Catastrophic health expenditures was defined in the variable specifications of materials and methods section in the text, line x, page x. Now we have replaced the abbreviation “CHE” with complete words” catastrophic health expenditures” to make it clearer.
“According to the WHO's definition, a household occurs catastrophic health expenditures when OOP spending on healthcare equaled or exceeded 40% of a household’s capacity to pay (CTP). CTP was defined as the total expenditure of the household minus the food-based household expenditure.”
- The conclusion and discussion part of the manuscript does not specify how policy makers and practitioners can use these research results. There are also no specific policy recommendations. The results do not make the necessary discussions about providing health services for the elderly and the prevention of catastrophic health expenditure.
A: Thanks for your comments. We have rewritten the conclusion part in the text, page 2-3, line 367-379 as shown below.
“Our study found the gap between different insurance types is narrowing. While UEMI still had an advantage, different insurance could be effective in promoting health service utilization and lowering OOP costs. The equity in healthcare utilization improved and healthcare costs had been controlled among middle-aged and elderly adults, but health insurance did not protect against CHE risks. Our study had several suggestions for policymakers. First of all, the government should actively promote hierarchical medical system and primary care system, and reasonably allocate medical resources to avoid waste. Second, insurance integration has progressed, but it should continue to narrow the gap between different types of insurances. Reimbursement for outpatient services should be increased to reduce the waste of inpatient resources. Third, health insurance could not reduce the risk of CHE, but is mainly related to patients’ financial situation and health status. The health insurance scheme should be tilted to socio-economic and poor-health groups and improve protection measures.”
- The discussion section needs to be rewritten by comparing the research results with similar international studies. Looking at the references section, many of the references belong to China. It is proposed to enrich the discussion part by using other studies.
A: Thank you for pointing this out. We have added similar international studies in the discussion section, in the text, page 1-2, line 309-353 as shown below.
“Our findings contrasted with evidence from previous studies which showed having health insurance may lead to higher total medical costs[6,10]. We only found an increase in total outpatient costs due to UEMI. This might have something to do with the fact that UEMI has higher reimbursement rate. The same situation has been reported in other countries, such as Germany[29] and Thailand[30], where the most generous social health insurance tends to incur the highest charges for the same types of medical conditions. It has been found that insurance has no financial protection for households. Some previous studies found that insurance could not reduce or even increase medical OOP costs[10,31]. However, some recent studies found that health insurance have been associated with boosting healthcare utilization and lower medical OOP costs[32-34], and our study supported the results. In detail, regarding inpatient costs, a study from India[34] found that social health insurance was significantly associated with decreasing inpatient OOP expenditures, consistent with our findings. As for outpatient costs, we concluded that social health insurance had little association with outpatient OOP expenditures, but conclusions in Turkey[35] and Ghana[36] were opposite. In recent years, Chinese medical reform policies have mainly focused on the reform of medical service prices, the encouragement of hierarchical diagnosis and treatment, and the encouragement of social medical services. China has launched a series of measures to control the unreasonable increase in medical costs[37,38]. The health insurance reform is achieved in the breadth of coverage in the population, the comprehensiveness of the benefits packages and increased reimbursement rates[39]. Our results may suggest that as Chinese health insurance system continues to improve, the equity in healthcare utilization has improved and health care costs have been effectively controlled. Health insurance’s induced spending effect due to the profit-driven nature of providers eased. This required more up-to-date research to support this conclusion.
A multi-country analysis showed most developed countries had advanced social institutions that protect households from CHE. Only the USA, Greece, Switzerland, and Portugal had more than 0.5% of households facing CHE[40]. The gap between China and developed countries was still quite large. Our results denoted that health insurance did not reduce the risk of CHE after adjusting for covariates, which was consistent with previous research[41-43]. China is still a developing country with a large proportion of low-income groups, which leaving social health insurance with little protection against CHE[44]. However, the findings differ from some of international studies that have shown that health insurance reduces the risk of CHE, depending on a number of factors, including different definitions of CHE[45,46], different thresholds[47-49], and differences in national circumstances[40]. Although health insurance could promote health services utilization by reducing OOP costs, the protective effect had been offset by the rapidly rising medical expenses and healthcare needs. What’s more, the reimbursement rates of health insurance were still inadequate for households with higher health service needs and lower economic levels[50]. As for the variables included according to the Andersen’s behavioral model, we also found participants with economically disadvantaged and poor health were more likely to have CHE[12,42,43]. It suggested that the policy of health insurance should be further improved to accurately identify the characteristics of the poor and incline toward the lower socio-economic and poor-health groups[43].”
Reviewer 2 Report
It is not recommended to quote data from the internet in a research paper ( 1 )
Scientific research comes from 2018, taking into account the period of the pandemic, certainly the costs incurred on health had different values ( health insurance and CHE risk). It,s worth extending the research to 2019-2020.
Author Response
- It is not recommended to quote data from the internet in a research paper ( 1 )
A: Thank you for pointing this out. We have changed the reference sources from the internet to the report and article.
“1. National Bureau of Statistics of China. Seventh national census. 2021.”
- Scientific research comes from 2018, taking into account the period of the pandemic, certainly the costs incurred on health had different values (health insurance and CHE risk). It’s worth extending the research to 2019-2020.
A: Thank you for your comments. The epidemic imposed many limitations on people's health service utilization and causes differences in health service outcomes. We strongly agree that is very important, and will further explore this research direction when the CHARLS data is updated. Now the most recent CHARLS data is till 2018.
Reviewer 3 Report
This paper studies the impact/correlation of different types of health insurance on/with healthcare utilization, costs, and catastrophic health expenditures (CHE) among the middle-aged and elderly in China. The authors concluded that all types of health insurance promoted hospital admissions and lowered out-of-pocket costs (OOP), but did not reduce the risk of CHE. They also found that equity in healthcare utilization was improved.
Given the skyrocket of healthcare expenditures in China and in other countries, the questions posed here have important policy implications. I have some comments as follows.
- Setting
I think before comparing the effects of various types of insurance, providing a comparison of their characteristics (for example, their coverage target, benefits, eligibility…) would help readers understand more about why we want to compare them, and understands the differences across the types of health insurance.
- Method
I think the models used in this paper allow the authors to make conclusions about the association between health insurance and healthcare behaviors and healthcare expenditures. There is not enough evidence to evaluate the causal impact of health insurance on the outcomes of interest. The causal effects between health insurance and healthcare utilization and healthcare costs can be both ways. People (sicker people) who anticipate that they might have a higher probability of using healthcare services and incurring higher costs are more likely to buy health insurance, so we usually observe a positive correlation between having health insurance and using healthcare services/having higher healthcare cost, which is also documented by the authors in this paper. To be able to evaluate causal impact, we need either randomized control trial or a good quasi-experimental research design. Thus, I recommend that the authors to use a different language when describing the results. The results estimated using the two-part models and OLS here reflect the correlation between those factors, not “effects” nor “impacts” of health insurance.
One of the objectives of the paper is to compare the correlation between health insurance and the outcomes of interest across different types of health insurance. Multinomial logit models are usually used in such circumstances and are considered a better approach than the two-part models in comparing multiple scenarios.
Also, can the authors write the model specifications in the method section?
- Minor points
- Please provide sources for Anderson’s behavior model (Section 2.2.3)
- What are the p-values reported in Tables 1 and 2? Are there any tests conducted there that I might have missed?
- The authors wrote that their results suggest that the equity in healthcare utilization has improved. Did the authors mean equity in what aspects of healthcare utilization (access, quality, cost)? I am not sure what result this conclusion is based on. I don’t see any evidence of improved equity from the results. Did the authors compare healthcare utilization across different income groups? (Section 4)
Reviewer 4 Report
Dear Authors,
It is very interesting topic. The research was designed in a logic and good well. The results are clearly presented. Data and method are presented well.
You could only mention about such research which were made in other countries.
Author Response
It is very interesting topic. The research was designed in a logic and good well. The results are clearly presented. Data and method are presented well. You could only mention about such research which were made in other countries.
A: Thank you for your comments. We have added similar international studies in the discussion section, in the text, page 1-2, line 309-353 as shown below.
“Our findings contrasted with evidence from previous studies which showed having health insurance may lead to higher total medical costs[6,10]. We only found an increase in total outpatient costs due to UEMI. This might have something to do with the fact that UEMI has higher reimbursement rate. The same situation has been reported in other countries, such as Germany[29] and Thailand[30], where the most generous social health insurance tends to incur the highest charges for the same types of medical conditions. It has been found that insurance has no financial protection for households. Some previous studies found that insurance could not reduce or even increase medical OOP costs[10,31]. However, some recent studies found that health insurance have been associated with boosting healthcare utilization and lower medical OOP costs[32-34], and our study supported the results. In detail, regarding inpatient costs, a study from India[34] found that social health insurance was significantly associated with decreasing inpatient OOP expenditures, consistent with our findings. As for outpatient costs, we concluded that social health insurance had little association with outpatient OOP expenditures, but conclusions in Turkey[35] and Ghana[36] were opposite. In recent years, Chinese medical reform policies have mainly focused on the reform of medical service prices, the encouragement of hierarchical diagnosis and treatment, and the encouragement of social medical services. China has launched a series of measures to control the unreasonable increase in medical costs[37,38]. The health insurance reform is achieved in the breadth of coverage in the population, the comprehensiveness of the benefits packages and increased reimbursement rates[39]. Our results may suggest that as Chinese health insurance system continues to improve, the equity in healthcare utilization has improved and health care costs have been effectively controlled. Health insurance’s induced spending effect due to the profit-driven nature of providers eased. This required more up-to-date research to support this conclusion.
A multi-country analysis showed most developed countries had advanced social institutions that protect households from CHE. Only the USA, Greece, Switzerland, and Portugal had more than 0.5% of households facing CHE[40]. The gap between China and developed countries was still quite large. Our results denoted that health insurance did not reduce the risk of CHE after adjusting for covariates, which was consistent with previous research[41-43]. China is still a developing country with a large proportion of low-income groups, which leaving social health insurance with little protection against CHE[44]. However, the findings differ from some of international studies that have shown that health insurance reduces the risk of CHE, depending on a number of factors, including different definitions of CHE[45,46], different thresholds[47-49], and differences in national circumstances[40]. Although health insurance could promote health services utilization by reducing OOP costs, the protective effect had been offset by the rapidly rising medical expenses and healthcare needs. What’s more, the reimbursement rates of health insurance were still inadequate for households with higher health service needs and lower economic levels[50]. As for the variables included according to the Andersen’s behavioral model, we also found participants with economically disadvantaged and poor health were more likely to have CHE[12,42,43]. It suggested that the policy of health insurance should be further improved to accurately identify the characteristics of the poor and incline toward the lower socio-economic and poor-health groups[43].”